# AAV-Mediated Gene Delivery to 3D Retinal Organoids Derived from Human Induced Pluripotent Stem Cells

**DOI:** 10.3390/ijms21030994

**Published:** 2020-02-03

**Authors:** Marcela Garita-Hernandez, Fiona Routet, Laure Guibbal, Hanen Khabou, Lyes Toualbi, Luisa Riancho, Sacha Reichman, Jens Duebel, Jose-Alain Sahel, Olivier Goureau, Deniz Dalkara

**Affiliations:** 1Sorbonne Université,, INSERM, CNRS, Institut de la Vision, 17 rue Moreau, F-75012 Paris, France; marcela.garita@inserm.fr (M.G.-H.); routetfiona@yahoo.fr (F.R.); laure.guibbal@gmail.com (L.G.); h.khabou@gmail.com (H.K.); ltoualbi@gmail.com (L.T.); luisa.riancho@inserm.fr (L.R.); sacha.reichman@inserm.fr (S.R.); jens.duebel@gmail.com (J.D.); j.sahel@gmail.com (J.-A.S.); olivier.goureau@inserm.fr (O.G.); 2Institut de Neurosciences de Montpellier, Université de Montpellier, INSERM, 34090 Montpellier, France; 3Department of Ophthalmology, University Medical Center Göttingen, 37075 Göttingen, Germany; 4CHNO des Quinze−Vingts, DHU Sight Restore, INSERM-DGOS CIC 1423, 75012 Paris, France; 5Department of Ophthalmology, The University of Pittsburgh School of Medicine, Pittsburgh, PA 15106, USA

**Keywords:** retinal organoids, hiPS cells, RPE, gene delivery, AAV

## Abstract

Human induced pluripotent stem cells (hiPSCs) promise a great number of future applications to investigate retinal development, pathophysiology and cell therapies for retinal degenerative diseases. Specific approaches to genetically modulate hiPSC would be valuable for all of these applications. Vectors based on adeno-associated virus (AAV) have shown the ability for gene delivery to retinal organoids derived from hiPSCs. Thus far, little work has been carried out to investigate mechanisms of AAV-mediated gene delivery and the potential advantages of engineered AAVs to genetically modify retinal organoids. In this study, we compared the early transduction efficiency of several recombinant and engineered AAVs in hiPSC-derived RPE cells and retinal organoids in relation to the availability of their cell-surface receptors and as a function of time. The genetic variant AAV2-7m8 had a superior transduction efficiency when applied at day 44 of differentiation on retinal organoids and provided long-lasting expressions for at least 4 weeks after infection without compromising cell viability. All of the capsids we tested transduced the hiPSC-RPE cells, with the AAV2-7m8 variant being the most efficient. Transduction efficiency was correlated with the presence of primary cell-surface receptors on the hiPS-derived organoids. Our study explores some of the mechanisms of cell attachment of AAVs and reports long-term gene expression resulting from gene delivery in retinal organoids.

## 1. Introduction

hiPSCs recapitulate different aspects of retinal development contributing to our understanding of normal retinogenesis and providing opportunities for in vitro disease modelling. Today, optic cups, optic vesicles, photoreceptors (PRs) [1,2,3,4] and retinal pigment epithelium (RPE) cells [5,6,7] can be derived from hiPSCs. Most inherited retinal diseases affect the outer retina and to correct the disease phenotype, the delivery of therapeutic genes to hiPSCs-derived PRs and RPE cells constitutes an important step. Recombinant AAVs (rAAVs) are the first choice for gene delivery to retinal cells for gene therapy [8]. They are non-pathogenic, non-integrative, and highly efficient as they can infect post-mitotic cells and provide long-term expression of a given transgene [9]. There are over one hundred naturally occurring AAV variants and ten distinct rAAV serotypes differing in their natural tropism towards cells [10]. rAAV tropism is dictated by different primary and secondary receptors located on the cell surface allowing rAAV-cell attachment and entry. Following the binding, rAAVs are internalized by endocytosis and travel towards the nucleus via distinct pathways (Figure 1). Because of the localization of these receptors on target cells, the entry of certain rAAVs is facilitated over others. In the retina, the cell-type specificity also depends on the injection route with intravitreal injections, bringing the rAAVs in contact with inner retinal cells and subretinal injections and facilitating vector access to the outer retinal cells [8,11]. In our hiPSC-derived organoids [12,13], laminated neural retina obtained after several weeks of differentiation displays photoreceptor precursors on the outside and ganglion cell precursors in the center of the structures. In these organoids, the medium containing the rAAVs is first and foremost in contact with the photoreceptor precursors, mimicking subretinal injections. Subretinally, rAAV2 is used in several ongoing clinical trials aiming to improve or correct RPE function (NCT03602820, NCT00999609, NCT03597399, NCT02781480, NCT00643747). On the other hand, rAAV2 has been shown to be less efficient than rAAV8 and 9 for photoreceptor transduction in non-human primates [14,15]. More recently, a peptide insertion variant of rAAV2 with increased photoreceptor transduction in both mice and primate species has been developed [16,17]. Although rAAV tropism in non-human primates is expected to be close to the outcome on human retinal cells, less work has been invested in testing rAAV tropism on human retinal cells derived from hiPSCs [18,19,20,21].

In this work, we investigated the mechanisms by which different rAAV variants transduce retinal organoids and RPE derived from hiPSCs. To this aim, we first examined the transduction profile of AAV2-7m8 in relation to its parental serotype rAAV2 and other naturally occurring capsids frequently used for outer retinal gene delivery, such as rAAV8 and rAAV9. We found the capsid variants carrying the 7m8 peptide to be more efficient than their parental capsid serotypes, leading to high-level transduction of both photoreceptor precursors and RPE. We next assessed the mechanisms behind the different transduction efficacies obtained with each serotype by labelling their primary cell surface, universal and glycan receptors as well as their known co-receptors within hiPSC derived retinal organoids. The recently discovered AAVR (KIAA0319L), recognized as a universal AAV receptor, was also included in the present study [22]. We found a positive correlation between the presence of the receptors and rAAV infectivity within the organoids. Altogether, we believe that our findings will be useful to guide the choice of gene delivery vectors to transduce hiPSC derived retinal organoids and predict outcomes in human gene therapy using various engineered and naturally occurring rAAVs.

## 2. Results

### 2.1. Presence of the Universal AAV Receptor AAVR in 3D Retinal Organoids

Based on a previously published protocol [27], we generated cone-enriched retinal organoids [12]. After 30 days of culture, self-forming organoids were mechanically isolated and placed in 3D conditions. At this stage, the organoids presented a neural-retina like structure, giving rise to all the different retinal neuronal cell types. rAAVs were applied on day 44, corresponding to the onset of cone differentiation, as we have shown by upregulation of human cone arrestin by a qPCR time course analysis [13]. Cell cycle arrest due to Notch signaling inhibition was achieved by addition of DAPT, a selective gamma secretase inhibitor added for a week, 2 days prior to infection. As a direct consequence of this inhibition, we observe the loss of the neural-retina-like structure after day 49 (Figure 2A). For the present study, four different capsids with photoreceptor tropism were tested prior to this stage. rAAVs were delivered directly into the culture medium surrounding the organoids.

We first analyzed the distribution of the universal AAV receptor AAVR (KIAA0319L) in these structures as a preliminary predictor of rAAV expression patterns in comparison with mouse and human retinas (Figure 2B–G). Cryosections were prepared and confocal images were acquired after labeling with antibodies against KIAA0319L (red) and nuclear staining with DAPI (blue). We observed that AAVR is most strongly expressed in the center of the rosettes at day 44 (Appendix A) and in the outer layers of our cone-enriched retinal organoids at day 70, were photoreceptor cells are found at each timepoint of differentiation [12,13]. This finding is consistent with the expression pattern in mature human and mouse retinas (Figure 2B–G). The KIAA0319L staining was more abundant of day 70 organoids compared to day 44 corresponding to a greater number of cones (Figure 2D–G). We did not see any differences in AAVR distribution or expression levels between the different organoids, excluding this as a factor contributing to variability of expression using different AAV serotypes and genetic variants.

### 2.2. Tropism of Different rAAVs towards Human Retinal Organoids

Previous studies reported rAAV transduction of ESC-derived retinal organoids and iPSC-derived RPE using naturally occurring AAV serotypes 2, 5, 8 and 9 at later stages of differentiation [19]. In order to determine which AAV capsid is most efficient for gene delivery into retinal organoids at an earlier time point, we performed infections with recombinant capsids AAV2, -8 and -9, alongside peptide insertion variant of AAV2 (AAV2-7m8). All of the vectors expressed GFP under the control of the ubiquitous promoter CAG. Similarly, to previous studies, we used a dose of 5 × 10^10^ vg per organoid. Live confocal imaging at day 70 of differentiation showed that the AAV2-7m8 variant led to the highest GFP expression on the surface of the organoids (Figure 3A–D). To quantitatively compare the percentage of GFP positive cells, we conducted an FACS analysis of whole dissociated organoids. The AAV2-7m8 variant showed the best transduction efficiency with 64.4 ± 7.7% of GFP positive cells. AAV2-7m8 driven expression was followed in decreasing order by rAAV8 (6.8 ± 1.4%), rAAV9 (2.5 ± 0.4%). FACS data were normalized against the percentage of GFP-positive cells infected with rAAV2 (1.6 ± 1%) (Figure 3E; *p* < 0.05, Mann–Whitney Student’s *t*-test, *N* = three biological replicates) in all the cases an identical number of cells were gated and no effect on cell viability was observed with any serotype. Peak expression was observed within two weeks of injection in retinal organoids using AAV2-7m8 (Appendix A).

To characterize potential dose-dependent effects with the serotypes leading to lower transduction efficiencies, we performed infections at a dose of 5 × 10^11^ vg per organoid (Appendix A). There was no difference in gene delivery efficiency; ruling out that certain AAV capsids might be efficient only at higher doses. Increasing the dose was enough to note a higher GFP expression driven by rAAV2 and 9 in some sparse cells within the retinal organoids (Appendix A) but overall efficacy did not improve by using higher dosage (Appendix A).

### 2.3. AAV Transduction Efficiency Correlates with the Presence of Cell-Surface Receptors

In retinal organoids, the most efficient capsid was AAV2-7m8 (Figure 3). To explain this observation, we asked which AAV cell surface receptors and co-receptors were found in the retinal organoids at the timepoint of infection. Using immunohistochemistry and RT-PCR, we looked at the expression of the different AAV receptors and co-receptors within retinal organoids at the day of infection (day 44) and at day 70 of differentiation (Figure 4). Heparan sulfate proteoglycan (HSPG) is the primary cell surface attachment receptor for the rAAV2 and 2-7m8 capsid variants. Members of the family of HSPG, also known as N-Syndecans, are most abundantly found in neural tissue and are involved in the formation of retinal neural networks [28,29]. We therefore investigated the presence of these syndecans (namely syndecan-3), as well as the co-receptor Laminin Receptor 1, in day 44 retinal organoids in relation to GFP expression patterns observed with AAV2-7m8 (Figure 4A). In addition to Laminin receptor 1 (encoded by the RPSA gene), FGFR1 also acts as co-receptor for rAAV2 and likely the AAV2-7m8 capsid. The expression of both co-receptors was established by RT-PCR experiments (Figure 4B). Syndecans were found around the rosettes where the photoreceptor precursors are normally found (Figure 4C,D and Appendix A). These rosettes disappeared after the addition of DAPT, a gamma secretase inhibitor that selectively blocks the Notch signaling pathway and promotes the organization of photoreceptors into a layer in the outer part of the organoid. Syndecan and Laminin Receptor 1 expression was sustained at least until Day 70 (Figure 4C,D). The abundant expression of its receptors likely contributes to the transduction efficacy observed with AAV2-7m8 in retinal organoids. On the other hand, despite the presence of its receptors, AAV2 was a lot less efficacious for the transduction of retinal organoids. This might be explained by the higher rate of cell entry observed by AAV2-7m8 compared to its parental serotype AAV2 [17].

We could not investigate the reasons behind rAAV8 inefficiency in our organoid system, as its primary receptor remains unknown. We could not determine, therefore, whether its absence is the reason for the lack of expression observed with this capsid but it is clear that the sole presence of its co-receptor, Laminin Receptor 1, is not enough to produce successful infection of retinal organoids.

### 2.4. rAAV9 and Cone Transduction Efficiency

rAAV9 has successfully been used to deliver genes to cones in non-human primates [30,31] and in mice thanks to the abundant presence of its primary receptor N-linked galactose in cone cells (Figure 5A). Surprisingly, in our experiments, AAV9 showed a low transduction efficiency towards our cone-enriched human retinal organoids. This observation was corroborated with the undetectable presence of N-linked galactose in these organoids at the time of infection (data not shown) and at later timepoints up to day 70 (Figure 5B). We hypothesized that this lack of N-linked galactose might be due to the lack of outer segment structures where this glycan is normally found in mature retinas (Figure 5A). We therefore examined iPSC derived retinal organoids at a much later stage of 226 days of differentiation. Indeed, PNA lectin staining revealed abundant N-linked galactose in hiPSC-derived retinal organoids at this advanced differentiation stage, suggesting that this glycan is expressed only as the retinal photoreceptors mature, even in the absence of outer segments (Figure 5B,C). It is interesting to note that PNA lectin is confined to the cone cell body in the absence of outer segment formation in these 226 day-old organoids. The absence of GFP expression in retinal organoids at Day 70 is thus explained by the absence of rAAV9’s main receptor at this early stage.

### 2.5. AAV Mediated Transduction of hiPSC-Derived RPE

Our 3D retinal organoids consist in neural retina. In order to determine the most efficient AAV capsids in RPE cells derived from hiPSC (hi-RPE), we first differentiated RPE cells using a previously established protocol [27]. hi-RPE cells were fully characterized for the major markers of RPE lineage using specific markers (Appendix A). By D63 of differentiation, hi-RPE cells exhibited the typical cobblestone shape and were immunoreactive for the RPE-specific transcription factor MITF, and the tight and adherens junction marker ZO-1. Additionally, the phagocytosis marker, FAK, was detected (Appendix A). We optimized the AAV-mediated transduction of hi-RPE cells at passage 2 (P2) and passage 3 (P3). A semi-automated image analysis system was used to determine the percentage of GFP immunoreactive cells from digital images after the transduction (Figure 6A–D). Four different AAV capsids were tested, with the AAV2-7m8 variant achieving a higher rate of transduction in comparison with rAAV2, 8 and 9 (Figure 6E).

## 3. Discussion

Widespread utility of AAV in clinical studies and the distinct transduction profiles of various AAV serotypes in different species stresses the importance of evaluating AAV vector efficiency in a more relevant human model. Human retinal organoids derived from hiPSC have proven useful not only for cell replacement strategies, but also for drug screening, disease modelling, gene correction and recently, optogenetic screening and subcellular trafficking [12,32]. Moreover, with genome editing gaining momentum as a therapeutic modality, gene-editing tools also require testing in human organoids. To this end, we tested the ability of different AAVs in the mini human retinas.

AAVs, like most viruses, require a primary cell surface receptor as well as the presence of co-receptors for optimal attachment and cell entry. We thus judged it useful to evaluate the ability of different rAAVs and engineered variants to infect retinal organoids in relation to the presence of their receptors. To our knowledge, this is the first time different AAV receptors and co-receptors have been identified in human retinal organoids and its expression has been correlated with vector transduction efficiency. Since previous studies examined AAV-mediated transduction of mature organoids, we focused primarily on early days of differentiation. This study in early differentiation is necessary since it has been previously reported that the developmental stage of the retina and the presence of outer segments is critical for AAV-mediated transduction in photoreceptors in vivo [33,34]. Early transduction of organoids can be useful when genetic modifications are necessary in genes expressed early in retinal development, to study long-term expression of a particular transgene in vitro or when transplantation of a precursor population is necessary to promote cell integration in a host retina.

We hypothesized that iPS cell derived retinal organoids are likely to have varying amounts of membrane receptors for different AAV serotypes, leading to distinct transduction profiles with these AAVs. Furthermore, the presence of membrane receptors might vary with the differentiation protocol used, playing a crucial role in the rate of infection. In light of these, we first started by evaluating the presence of the universal AAV receptor AAVR in 44 day-old organoids. At this stage, the organoids are immature with early photoreceptors forming rosettes where AAVR was detected. Prolonged culture of the organoids allowed the formation of an outer layer enriched in cones at day 70, where AAVR was present in similar concentrations as those observed in mature mouse and human retinas. However, its presence was not determinant for the AAV-mediated transduction with some capsids, so AAVR is possibly more of a co-receptor or, as it was suggested before, playing a role in the trafficking of the virus once it has been internalized by the host cell [35].

This initial observation encouraged us to investigate frequently used rAAVs and their tropism on early-stage retinal organoids. We focused on the serotypes that are most frequently used for photoreceptor transduction in gene therapy studies. rAAV2, and the peptide insertion variant AAV2-7m8, rAAV8 and 9 were tested for their ability to drive reporter gene expression in retinal organoids when applied at day 44. We allowed for lengthy incubation times in order to rule out any differences in expression patterns due to individual capsid’s expression kinetics. We observed noticeable reporter gene expression using all AAVs in the outer parts of the retinal organoids. However, expression was significantly stronger using the engineered variant AAV2-7m8 compared to naturally occurring rAAVs.

In an effort to better understand the differences between AAV capsids’ performance, we examined the abundance of their known receptors within retinal organoids at the time of infection. In the case of rAAV2-mediated transduction, we could not establish a direct correlation between the presence of receptor Syndecan 3 and its low efficiency to infect retinal organoids. We hypothesize than other members of the HSPG family, different than Syndecan 3 are the primary receptor for rAAV2 and are absent in our organoids. AAV2-7m8 was more efficient in gene delivery to retinal organoids despite the fact that it shares common cell surface receptors with its parental capsid AAV2 [16]. AAV2-7m8 is more infectious than rAAV2, possibly due to the more efficient activation of the secondary receptor or binding to other receptors that have not yet been described. Moreover, other factors than the presence of the receptors can enter into play in culture conditions, such as the presence of growth factors in the medium. For example, the AAV2 co-receptor FGFR1 can be blocked by FGF2, which is used in the generation for both retinal organoids and RPE cells derived from hiPSCs. Infections with rAAV close to the administration of FGF2 can impede cell entry and reduce transduction efficacy with this serotype. Our limited knowledge on the cell surface receptors of AAV8 did not allow us to directly assess infectivity with this serotype in relation to cell surface receptors. Nevertheless, we found limited transduction of the retinal organoids by this serotype, which is known to be very efficient in vivo in several animal models. Our work is not the first one showing differences with in vivo results in several animal models, highlighting the need to validate AAV-mediated gene transfer in a relevant human retinal model such as the human retinas derived from hiPSC. Similarly, rAAV9 gave rise to poor transduction in our retinal organoids despite being highly successful in targeting cones in mice and nonhuman primates [25,30]. This observation can be partially explained by the absence of N-linked-galactose, AAV9’s primary receptor, in our retinal organoids at the time of infection and at the time of analysis. N-linked galactose is found in the outer segments of cones in mature human and mouse retinas. Our data suggest that N-linked galactose is only present at much later stages of differentiation in retinal organoids. The time at which N-linked galactose is present may vary with the differentiation protocol and the iPSC line used, but it could explain why targeting organoids with AAV9 has been challenging in the past [18,19].

## 4. Materials and Methods

### 4.1. Animals

The wild-type C57BL/6 mice (Janvier Laboratories) used in this work were 7 to 8 weeks old at the time of infection and immunohistochemical analysis was performed 4 weeks after. All mice were housed under a 12-hour light-dark cycle with water and food at libitum and all procedures were approved by the local animal experimentation ethics committee (Le Comité d’Ethique pour l’Expérimentation Animale Charles Darwin) and were carried out according to institutional guidelines in adherence with the National Institutes of Health guide for the care and use of laboratory animals as well as the Directive 2010/63/EU of the European Parliament.

### 4.2. Maintenance of hiPSC Culture

All the experiments conducted in this study were carried out using the hiPSC-2 cell line, previously established from human fibroblasts [3] and recently adapted to feeder-free conditions [27]. Cells were kept at 37 °C, under 5% CO_2_/95% air atmosphere_,_ 20% oxygen tension and 80%–85% humidity. Colonies were cultured with Essential 8™ medium (ThermoFisher scientific, Waltham, MA, USA) in culture dishes coated with truncated recombinant human vitronectin and passaged once a week, as previously described [27].

### 4.3. Differentiation of Human iPS Cells into Retinal Organoids

Optimization of previous protocols [3,27] allowed the generation of retinal organoids from human iPSC. In brief, hiPSC cell lines were expanded until 80% confluence in Essential 8™ medium before being switched to Essential 6™ medium (Thermo Fischer Scientific). After 3 days, cells were cultured in Proneural medium. At day 28, NR-like structures grew out of the cultures and were mechanically isolated and further cultured in a 3D system in Maturation medium until day 70 of differentiation. Table 1 summarizes media formulation. Floating organoids were passaged into 6-well plates (10 organoids per well) and supplemented with 10 ng/mL FGF2 (Peprotech, Rocky Hill, NJ, USA) until D35. Additionally, between day 42 and day 49, 10 µM DAPT (Selleckchem, Houston, TX, USA) was added to Maturation medium in order to promote the photoreceptor commitment of retinal progenitors. Medium was changed every 2–3 days (Figure 2).

### 4.4. Generation of RPE from Human iPS Cells (hi-RPE Cells)

RPE cells were generated under feeder-free conditions. After removal of neural retinal structures on day 28, the rest of the culture was kept in RPE medium (Table 1). Around day 42, emerging patches of pigmented cells were mechanically dissected and transferred into a Geltrex™ (Thermo Fisher Scientific)-coated 24-well plate; this is noted as passage P0. Density was kept at maximum of 10 patches per well in 1 mL of RPE medium. Upon confluence, hi-RPE cells were passaged using 0.25% Trypsin-EDTA (Thermo Fisher Scientific) for 10–15 min at 37 °C. Three confluent wells from the 24-well plate were passaged into a Geltrex™-coated T25 cm² Flask (BD VWR) with 5 mL or RPE medium (Passage P1). P1 hi-RPE cells were cryopreserved in CryoStem™ hPSC Freezing Medium (Biological Industries, Cromwell, CT, USA) at a cell density of 1.5 × 10^6^ cells per vial or used for experiments until Passage 3 (P3) via enzymatic cell dissociation (TrypLE; Thermo Fisher Scientific). In all passages, the medium was changed every 2–3 days and supplemented with 10 ng/mL FGF2 (Preprotech) for a week.

### 4.5. AAV Vector Production

Recombinant AAVs were produced as previously described [12] using the co-transfection method and purified by iodixanol gradient ultracentrifugation. Concentration and buffer exchange was performed against PBS containing 0.001% Pluronic. AAV vector stocks titers were then determined by the real-time quantitative PCR titration method using SYBR Green (Thermo Fischer Scientific).

### 4.6. Infection of Retinal Organoids with AAV Capsids

Direct addition to the medium was performed in 6-well plates, where each well contained a group of 10–12 organoids. Infections were done at day 44 at 5 × 10^10^ vg or 5 × 10^11^ vg per organoid directly added in the Proneural medium (Table 1). Medium was change 72 h after infection.

### 4.7. Infection of hiPSC-Derived RPE with AAV Capsids

RPE cells from passages P2 and P3, were plated in 12 mm coverslips coated with Geltrex™ (Thermo Fischer Scientific) at 70% confluence (230,000 cells per well) and infections were performed by direct addition of AAV solutions to the RPE medium (Table 1) at 5 × 10^9^ vg or 5 × 10^10^ vg. Medium was changed 72 h after infection. RPE transduction was examined 4 weeks after infection.

### 4.8. Subretinal Injections

Mice were anesthetized by isofluorane inhalation. Pupils were dilated with 2.5% phenylephrine and a 33-gauge needle was inserted into the eye to deliver 1 μL of AAV vector solution (10^10^ vg) into the subretinal space.

### 4.9. Tissue Preparation

Seventy day-old organoids were washed in PBS and fixed in 4% paraformaldehyde for 10 min at 4 °C before they were incubated overnight in 30% sucrose (Sigma-Aldrich, St. Louis, MO, USA) in PBS. Organoids were embedded in gelatin blocks (7.5% Gelatin -Sigma-Aldrich-, 10% Sucrose in PBS) and frozen using isopentane at −65 °C.

Mice eyes were fixed in formalin 2 h at RT. Eyes were dissected to obtain only the back part of the eye, with the retina and the RPE, before overnight incubation at 4 °C in 30% sucrose/PBS solution. Samples were included in O.C.T. (Avantor, Radnor, PA, USA) and frozen using liquid nitrogen and stored at −20 °C.

### 4.10. Cryosectionning and Immunohistochemistry

Ten um-thick organoid sections were obtained with a Cryostat Microm and mounted on Super Frost Ultra Plus^®^ slides (Menzel Gläser, Braunscheig, Germany). Samples were washed in PBS to remove the rest of O.C.T. and then permeabilized in PBS containing 0.5 % Triton® X-100 during 1 h at RT. Blocking was done with PBS containing 0.2% gelatin and 0.25% Triton X-100 for 30 min at RT. Incubation with primary antibodies (Table 2 was performed overnight at 4 °C. Several washes with PBS containing 0.25% Tween20 were performed before incubation with Fluorochrome-conjugated secondary antibodies (1/500 dilution) and DAPI (4’-6’-diamino-2-phenylindole, dilactate; Invitrogen-Molecular Probe; 1/1000 dilution) to counterstain the nuclei for 1 h at RT. Samples were further washed in PBS and dehydrated with 100% ethanol before mounting using fluoromount Vectashield (Vector Laboratories, Peterborough, UK).

### 4.11. Image Acquisition

Immunofluorescence was observed using a Leica DM6000 microscope (Leica microsystems, Wetzlar, Germany) equipped with a CCD CoolSNAP-HQ camera (Roper Scientific, Vianen, Netherlands) or using an Olympus FV1000 confocal microscope equipped with 405, 488, and 543-nm pulsing lasers. Confocal images were acquired using an optimized 1.55 or 0.46 µm step size based on the Nyquist-Shannon theorem and corresponded to the projection of 4–8 optical sections.

### 4.12. RNA Isolation and RT-PCR

Total RNA was extracted using a RNeasy mini kit (Qiagen, Düsseldorf, Germany) following the manufacturer’s instructions. RNA concentration and purity were determined using a NanoDrop ND-1000 Spectrophotometer (Thermo Scientific). Reverse transcription was carried out with 1 μg of total RNA using the QuantiTect retrotranscription kit (Qiagen). All the samples were normalized against a housekeeping gene (GAPDH). The primer pairs (IDT) are listed in Table 3 All RT-PCR reactions were run for 40 cycles of 94 °C for 20 s, 58 °C for 1 min and 68 °C for 30 s.

### 4.13. Cell Counts on RPE Cells

For the image analysis, at least 12 microscopic fields from each sample were taken randomly using a 40X lens objective in an Olympus FV1000 confocal microscope. To reduce human bias, a semi-automated image analysis system was used to determine the percentage of immunoreactive cells from digital images using Metamorph NX^®^ v7.5.1.0 Software (Molecular Devices, LLC, USA).

### 4.14. FACS Analysis

Quantification of GFP positive cells in retinal organoids was performed in 3 biological replicates, meaning that each sample of 10 organoids was obtained from 3 different protocols of differentiation. Each sample was dissociated using two units of pre-activated papain at 28.7 u/mg (Worthington) in Ringer solution for 25 min at 37 °C. Once a homogeneous cell suspension was obtained by repeated pipetting, papain was deactivated with Proneural medium (Table 1). Cells were filtered with 30 µm filter (Miltenyi, Bergisch Gladbach, Germany) and fixed for 10 min at 4 °C with 4% paraformaldehyde. Cells were washed in PBS. Samples were analysed by flow cytometry and at least 10,000 events were analysed in each experiment using FACSCalibur system (BD Biosciences, Allschwil, Switzerland). The number of positive cells within the gated population was analysed using CellQuest™ Pro (BD Biosciences) software. Non-infected organoids serve as controls.

### 4.15. Statistical Analyses

Data were analysed with GraphPad Prism and they are expressed as mean ± standard error of mean (SEM) of at least 3 independent biological replicates (*N* = 3), except for immunocytochemistry for which a representative image from at least 3 independent assays was depicted in the figures. Comparison between values were analyzed using unpaired two-tailed non-parametric Mann-Whitney Student’s test. *p* values are indicated in each figure, but a *p* < 0.05 was considered significant.

## 5. Conclusions

Retinal organoids and RPE cells derived from hiPSC are a unique tool to evaluate human gene therapy approaches in vitro. They provide a human and retinal context to predict the efficacy of gene expression driven by viral constructs and provide opportunities to test gene editing. Different AAVs present different efficacies depending of the type and number of cells within the retinal organoids, which depend on the differentiation protocol. This is the first time a mechanistic explanation for the diverse AAV-mediated expression in organoids is proposed based on the availability of cell surface receptors and co-receptors. AAV2-7m8 is the best capsid variant for efficient early transduction of our 3D retinal organoids as well as hi-RPE cells. The transduction levels in immature 3D retinal organoids differ from what is observed in vivo in primates and rodents, most likely due to the absence of AAVs cell surface receptors within the organoids. This must be studied further once more AAVs’ binding proteins are identified but it should be taken into consideration when gene delivery in a retinal organoid is needed. Monolayers of hiRPE cells are more efficiently transduced using the AAV serotypes tested in this study, potentially due to the better accessibility of the cell surface. Nevertheless, AAV2-7m8 is more efficient than the other AAVs tested under identical conditions, suggesting its utility for gene delivery to hi-RPE.

## Figures and Tables

**Figure 1 ijms-21-00994-f001:**
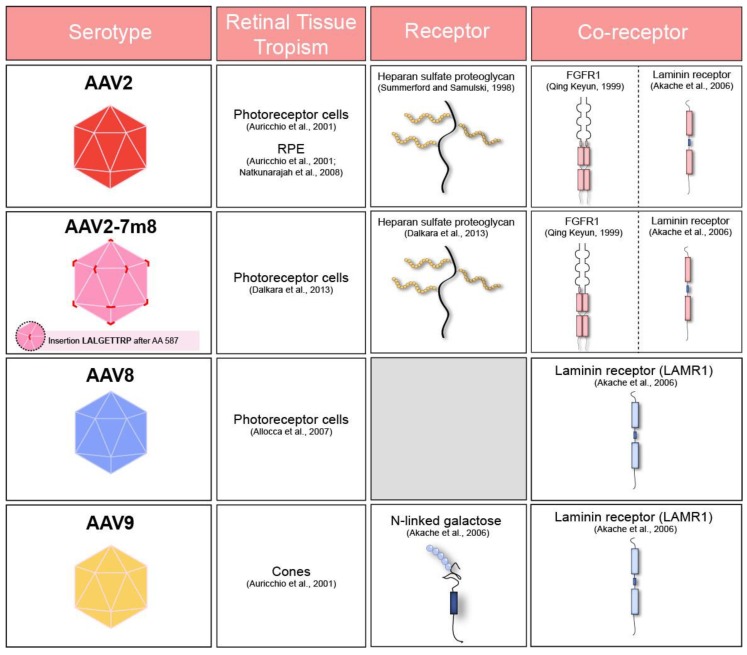
Table summarizing adeno-associated virus (AAV) receptors, co-receptors and cell tropism of commonly used serotypes after subretinal administration in murine [23,24,25] and primate retina [26]. FGFR1: Fibroblast Growth Factor Receptor 1. LAMR1: Laminin Receptor 1.

**Figure 2 ijms-21-00994-f002:**
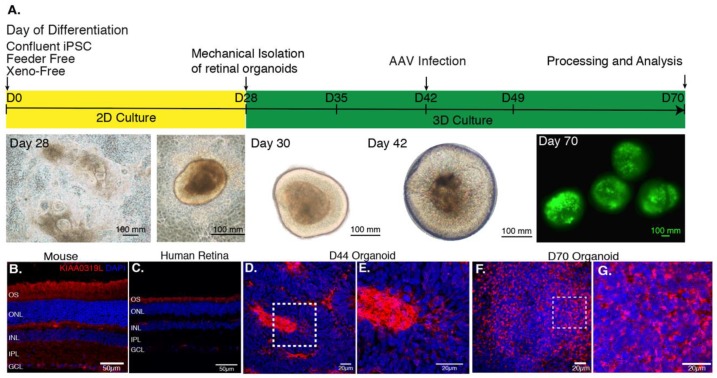
(**A**) AAV-mediated gene delivery to retinal organoids. Schematic of the protocol for generating and infecting human retinal organoids with representative images from each step. rAAV infection is performed at day 44 of differentiation. The efficiency of gene delivery illustrated on day 70 corresponds to one-single infection using AAV2-7m8-GFP at a dose of 5 × 10^10^ vg per organoid. (**B**) Expression of AAV receptor KIAA0319L in mature mouse retina, human retina and hiPSC-derived organoids. Representative retinal cross-section of wild-type mouse (**B**) or human retina (**C**) stained with antibodies against the KIAA0319L receptor. (**D**) Expression of AAVR KIAA0319L in Day 44 old hiPSC-derived organoids. (**E**) Zoom into the dotted square depicted in D. (**F**) Expression of AAVR KIAA0319L in day 70 old hiPSC-derived organoids. (**G).** Zoom into the dotted square depicted in F. Scale bars: A and D = 50 μm; B–C; E–F = 20 μm.

**Figure 3 ijms-21-00994-f003:**
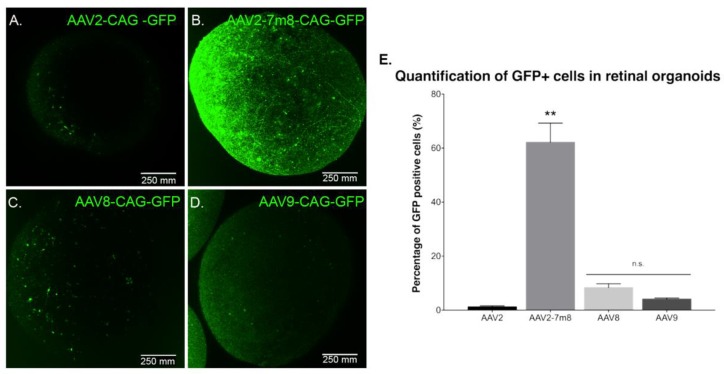
GFP expression as a function of the serotype. Live confocal imaging of representative retinal organoids showing GFP Expression driven by the CAG promoter for the four different tested capsids. (**A**) AAV2-CAG-GFP. (**B**) AAV2-7m8-CAG-GFP. (**C**) AAV8-CAG-GFP and (**D**) AAV9-CAG-GFP. In all cases, the infections were performed at a viral concentration of 5 × 10^10^ vg per organoid at day 44 of differentiation. Scale bar: 250 µm. (**E**) Percentage of GFP positive cells quantified by FACS analysis. *N* = 3 biological replicates of *n* = 10 organoids. Values are mean ± SEM. For statistical significance, Mann–Whitney Student’s test was used and ** *p* < 0.05 was considered significant. n.s. = non significant.

**Figure 4 ijms-21-00994-f004:**
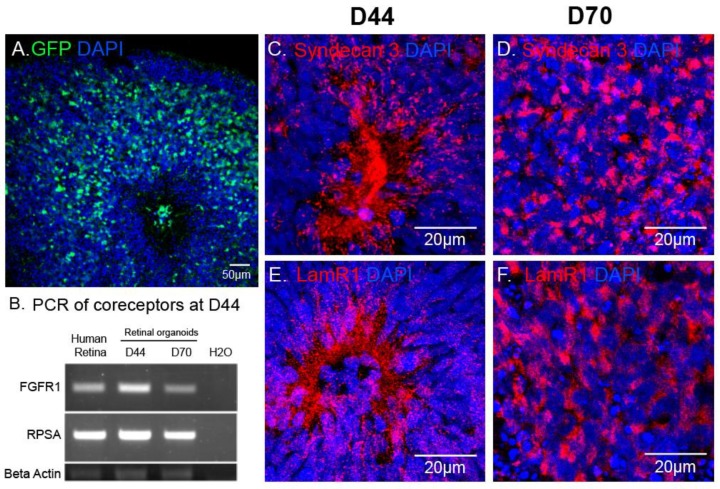
AAV mediated reporter gene expression in relation to the distribution of AAV receptors in hiPSC-derived retinal organoids. (**A**) Confocal image of a D70 retinal organoid infected with AAV2-7m8-CAG-GFP, GFP signal was amplified using chicken polyclonal anti-GFP antibody. (**B**) RT-PCR showing expression of co-receptor genes (Laminin receptor1 (RPSA) and FGFR1) in the mature human retina, D40 and D70 retinal organoids. (**C–F**) Confocal images showing localization of Syndecan 3 and Laminin receptor 1 in 3D retinal organoids at different stages of differentiation. Scale bars: A, 50 μm; C–H, 20 μm.

**Figure 5 ijms-21-00994-f005:**
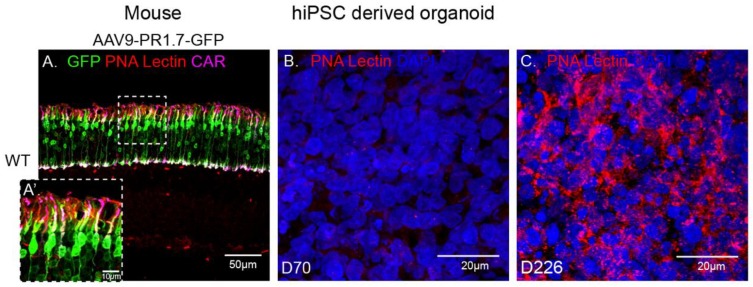
Immunofluorescence analysis of PNA lectin (red) in a mature wild-type mouse retina injected subretinally with AAV9-GFP, co-stained with cone arrestin (magenta) and DAPI (blue) (**A**) versus D70 (**B**) and D226 (**C**) hiPSC derived retinal organoids. Scale bar: 50 µm.

**Figure 6 ijms-21-00994-f006:**
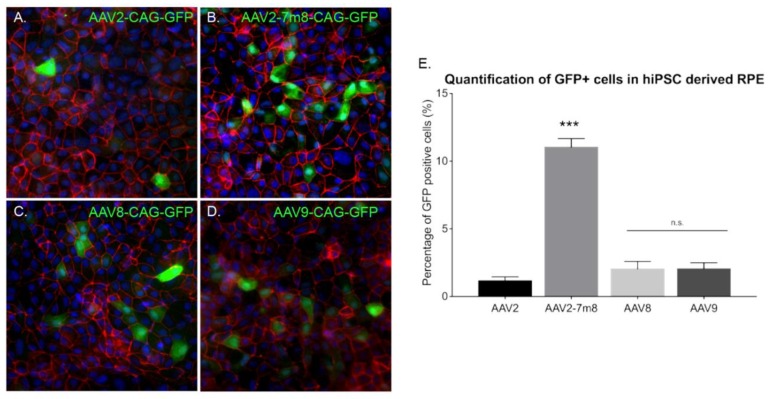
AAV mediated gene delivery to hiPSC derived RPE. (**A–D**) Epifluorescence images of the expression of the RPE structural marker ZO1 (red), the reporter protein GFP (green), and the nucleus marker DAPI (blue) in hiPSC-derived RPE cultures infected with different AAVs. Scale bar: 50 µm (**A**) AAV2-CAG-GFP, (**B**) AAV2-7m8-CAG-GFP, (**C**) AAV8-CAG-GFP, (**D**) AAV9-CAG-GFP. (**E**) Percentage (%) of GFP positive cells in hiPSC-derived RPE cultures infected with different capsids quantified using ImageJ analysis. *N* = 3. Values are mean ± SEM. For statistical significance Mann–Whitney Student’s test was used and ****p* < 0.05 was considered significant. n.s. was used to declare non significant.

**Table 1 ijms-21-00994-t001:** Media composition.

**Proneural Medium**	Essential 6 Medium (A1516401) N-2 supplement (100X) 1% (17502048) Penicillin-Streptomycin 1% (15140122)
**Maturation Medium**	DMEM/F-12 (11320074) B27 supplement (100X), 2% (17504044) MEM Non-Essential Amino Acid Solution (100X), 1% (11140035) Penicillin-Streptomycin 1% (15140122)
**RPE Medium**	DMEM/F-12 (11320074) N-2 supplement (100X) 1% (17502048) Penicillin-Streptomycin 1% (15140122)

**Table 2 ijms-21-00994-t002:** Primary Antibodies and Lectins.

Antigen	Manufacturer/Reference	Specie/Clonality	Dilution
GFP	Abcam ab13970	Chicken polyclonal	1/500
Syndecan3	Antibodies-online ABIN4352331	Rabbit polyclonal	1/200
RPSA (Laminin Receptor 1)	Abcam ab137388	Rabbit polyclonal	1/500
PNA Rhodamine conjugated	Vector Laboratories RL-1072	Lectin	1/100
KIAA0319L (AAVR)	Abcam ab105385	Mouse polyclonal	1/200
ZO-1	Invitrogen 61-7300	Rabbit polyclonal	1/2000
MITF	DAKO M3621	Mouse monoclonal	1/100
FAK	Millipore 05-185	Mouse monoclonal	1/100
Cone Arrestin	Millipore MAB15282	Rabbit polyclonal	1/2000
CRX	Abnova H00001406-M02	Mouse monoclonal	1/5000

**Table 3 ijms-21-00994-t003:** Primer sets.

**FGFR1**	Forward: CCAGACAACCTGCCTTATGT Reverse: GCTGTGGAAGTCACTCTTCTT Amplicon size 327 bp
**RPSA** (Laminin Receptor 1)	Forward: CACTCCTGGAACCTTCACTAAC Reverse: CTCCAGTCTTCAGTAGGGAATTG Amplicon size 495 bp

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
