# Peer review of "AAV-Mediated Gene Delivery to 3D Retinal Organoids Derived from Human Induced Pluripotent Stem Cells"

_ijms, 2020, doi:10.3390/ijms21030994_

Round 1

Reviewer 1 Report

Review of Manuscript “AAV-mediated gene delivery to 3D retinal organoids derived from human induced pluripotent stem cells” by Garita-Hernandez et al..

In their study the authors describe the transduction properties of recombinant vectors derived from AAV serotypes 2, 8 and 9 as well as of a previously described, genetically modified, variant of the AAV2 capsid with an insertion of a peptide at amino acid position 587 (AAV2-7m8) in hiPSC (human induced pluripotent stem cells)-derived retinal organoids and retinal pigment epithelium (RPE) cells. In parallel the availability of the corresponding surface receptors and co-receptors was determined. The major difference to very similar earlier studies (references 19 and 21) was the focus on early days of retinal cell differentiation.

In both organoids and RPE, superior transduction properties of the rAAV2-7m8 variant were found as compared to the vectors derived from wildtype AAV serotypes 2, 8 and 9. At least for rAAV-9 the reduced transduction on cone enriched retinal organoids could be correlated to the absence of N-linked galactose as primary receptor.

The study is well written and structured and the conclusions are supported by the experimental data. One of its major shortcomings is the lack of explanation for the vast difference in transduction efficiencies between the vectors derived from the wild AAV2 capsid and the peptide inserted rAAV2-7m8 variant, since both seem to use the same cellular receptors. The relative transduction efficiencies among the AAV serotypes also differ markedly from those obtained in vivo in several animal models. Therefore, it might have been favorable to include more serotypes in the study. Further major and minor point, which should be addressed in a revised version of the manuscript, are listed in detail below.

Detailed points of criticism:

Major points:

1) At present the discussion section is largely a repetition and summary of the results sections. Ideally, there should be a greater focus on possible implications of the experiments in view of the existing data and possible approaches to answer the open questions in future studies.

2) The authors generally use the term AAV to denote recombinant vectors based on adeno-associated virus. To avoid misunderstandings and clearly differentiate between the wildtyp AAVs and the recombinant vectors derived thereof, it would be preferable to use the term “rAAV“ for the recombinant AAVs.

3) The universal AAV receptor AAVR described by Pillay et al. constitutes a major part of the experiments shown in fig. 2, but is only first mentioned without further explanation in lines 102/103 in the results section. The properties of this receptor should be described in the introduction section.

Minor points:

1) Line 25: In the abstract the abbreviation AAV2-7m8 for the genetically modified rAAV2 variant is used without further explanation, which strongly limits the comprehensibility when reading the abstract only. Furthermore, this engineered variant should not be denoted as a serotype as done in line 28.    

2) Results section fig. 1: Although highly descriptive, fig. 1 is barely readable in a printout. Maybe part B of the figure could be omitted in favor of an enlargement of part A, since (B) does not contain any detailed information, which could not be given in one or two short statements in the text.

3) Major parts of Fig. S2 are highly over-exposed.

4) Fig. 5: The upper and lower parts of the figure seem to be identical.

5) Line 146 to 148: Check phrasing

6) Khabou et al. is cited by author and not by numbers as the other references.

Author Response

Response to Reviewer 1 Comments

Comments and Suggestions for Authors

Review of Manuscript “AAV-mediated gene delivery to 3D retinal organoids derived from human induced pluripotent stem cells” by Garita-Hernandez et al..

In their study the authors describe the transduction properties of recombinant vectors derived from AAV serotypes 2, 8 and 9 as well as of a previously described, genetically modified, variant of the AAV2 capsid with an insertion of a peptide at amino acid position 587 (AAV2-7m8) in hiPSC (human induced pluripotent stem cells)-derived retinal organoids and retinal pigment epithelium (RPE) cells. In parallel the availability of the corresponding surface receptors and co-receptors was determined. The major difference to very similar earlier studies (references 19 and 21) was the focus on early days of retinal cell differentiation.

In both organoids and RPE, superior transduction properties of the rAAV2-7m8 variant were found as compared to the vectors derived from wildtype AAV serotypes 2, 8 and 9. At least for rAAV-9 the reduced transduction on cone enriched retinal organoids could be correlated to the absence of N-linked galactose as primary receptor.

The study is well written and structured and the conclusions are supported by the experimental data. One of its major shortcomings is the lack of explanation for the vast difference in transduction efficiencies between the vectors derived from the wild AAV2 capsid and the peptide inserted rAAV2-7m8 variant, since both seem to use the same cellular receptors. The relative transduction efficiencies among the AAV serotypes also differ markedly from those obtained in vivo in several animal models. Therefore, it might have been favorable to include more serotypes in the study. Further major and minor point, which should be addressed in a revised version of the manuscript, are listed in detail below

We would like to thank reviewer 1 for the thorough revision of our work and for recognizing the difference of our work with previous studies and also for finding that our experimental data support our conclusions but above all for allowing us with his advice to improve our manuscript.

Significant changes have been made to the text of the manuscript in order to follow your recommendations and improve our explanations regarding:

The vast difference of AAV2 and AAV2-7m8 capsids Differences with in vivo results in several animal models

To the other points we respond in detail below.

Detailed points of criticism:

Major points:

1) At present the discussion section is largely a repetition and summary of the results sections. Ideally, there should be a greater focus on possible implications of the experiments in view of the existing data and possible approaches to answer the open questions in future studies.

Response 1. We have followed your recommendation and enlarge the discussion section. Now we are including more possible implications of our experiments and open questions to be addressed in future studies

2) The authors generally use the term AAV to denote recombinant vectors based on adeno-associated virus. To avoid misunderstandings and clearly differentiate between the wildtyp AAVs and the recombinant vectors derived thereof, it would be preferable to use the term “rAAV“ for the recombinant AAVs.

Response 2. We have replaced AAV for rAAV all along the manuscript

3) The universal AAV receptor AAVR described by Pillay et al. constitutes a major part of the experiments shown in fig. 2 but is only first mentioned without further explanation in lines 102/103 in the results section. The properties of this receptor should be described in the introduction section.

Response 3. Description of AAVR is now found in the introduction section as well, line 72

Minor points:

1) Line 25: In the abstract the abbreviation AAV2-7m8 for the genetically modified rAAV2 variant is used without further explanation, which strongly limits the comprehensibility when reading the abstract only. Furthermore, this engineered variant should not be denoted as a serotype as done in line 28.    

Response 1. AAV2-7m8 variant has been further explained in the abstract, line 25 to make sure is not confused with rAAV2. Also, the engineered variant is specified as such in line 28 and not as a serotype.

2) Results section fig. 1: Although highly descriptive, fig. 1 is barely readable in a printout. Maybe part B of the figure could be omitted in favor of an enlargement of part A, since (B) does not contain any detailed information, which could not be given in one or two short statements in the text.

Response 2. Figure 1B has been omitted and instead we have described the mechanism in the text in the introduction, line 47

3) Major parts of Fig. S2 are highly over-exposed.

Response 3. Figure S2 corresponds to the transduction kinetics for AAV2-7m8 along 4 weeks and for comparative reasons all images were acquired with the same settings. Over-exposed images are inevitable in order to see expression of the transgene at earlier time points of the analysis

4) Fig. 5: The upper and lower parts of the figure seem to be identical.

Response 4. By mistake Fig 5 was duplicated. This has been rectified in the revised manuscript

5) Line 146 to 148: Check phrasing

Response 5. Effectively the phrase needed some corrections and in the revised version the phrase reads: “Besides Laminin receptor 1 (encoded by the RPSA gene), FGFR1 also act as co-receptor for AAV2 and likely AAV2-7m8 capsid.” Lines: 155-157

6) Khabou et al. is cited by author and not by numbers as the other references.

Response 6. Thank you for highlighting this error. It was indeed a mistake in the reference database used and it has been rectified. Line 166

Reviewer 2 Report

The manuscript by Garita-Hernandez et al. describes AAV-mediated gene delivery to retinal organoids of different age. While the depiction of the results is nice, there is several concerns to the claims of the authors:

Line 152/3:
Expression of Lam and Syndecan at day70
I find no specific proof that there is a correlation between that and the 7m8 efficacy. Maybe a knockdown or inhibition would help to strengthen the claim.

Line 156:
One of the major aspects (AAV2 vs 7m8) is just refered to a hypothesis from a publication

Line 191:
AAV9
-->PNA lectin only present at day 226
Why not test infection at older stages? (Will also not work, so no proof for a correlation here)

Figure 3and 6:
RO and RPE transduction was already done by Gonzalez codero 2018 (Citation 19) exceptt for 7m8.
Please explain the novelty and impact

Figure 5: is doubled

Author Response

Response to Reviewer 2 Comments

The manuscript by Garita-Hernandez et al. describes AAV-mediated gene delivery to retinal organoids of different age. While the depiction of the results is nice, there is several concerns to the claims of the authors:

We appreciate the revision performed by Reviewer 2 and the opportunity to improve our manuscript and to put in value our results. We just want to clarify that out study focus in early transduction of retinal organoids and we have made some changes in the manuscript to highlight this as it was not sufficiently clear in the submitted version.

We respond in detail to the reviewer concerns below.

Line 152/3:
Expression of Lam and Syndecan at day70
I find no specific proof that there is a correlation between that and the 7m8 efficacy. Maybe a knockdown or inhibition would help to strengthen the claim.

Line 156:
One of the major aspects (AAV2 vs 7m8) is just refered to a hypothesis from a publication

Response 1 and 2. Indeed, we have not done any knockdown experiments to support this claim however, previous experiments already probed glycan dependencies of 7m8 compared to AAV2 (see supplementary figure 1 Dalkara et al., 2013). In this prior work ex vivo characterization of 7m8 on relevant cell types was performed compared to AAV2 on CHO, PgsA, Pro5, and Lec1 cells. CHO/PgsA transduction probes the HSPG dependence, whereas Pro5/Lec1 transduction examines the sialic acid dependence, of AAV2 and 7m8. Furthermore, heparin binding profiles of 7m8 and AAV2 were assessed by heparin column elution profile by chromatography, thus supporting experimentally the hypothesis from previous publication.

Line 191:
AAV9
-->PNA lectin only present at day 226
Why not test infection at older stages? (Will also not work, so no proof for a correlation here)

Response 3. The aim of this paper is to compare the efficiency of transduction by different AAV capsids early in differentiation. AAV9 receptor, N-galactose, identified by PNA lectin is not present in our organoids at the time of infection (D44) nor at the end of the experiments (D70). The absence of PNA implies this is the reason for the lack of expression observed with AAV9. However, we did show the presence of PNA in mature organoids (D226) using our protocol of differentiation, and we suggest the time at which PNA becomes available will vary with the protocol of differentiation used. Looking for the presence of the receptor in the organoids before attempting AAV-mediated transduction is thus proposed for readers attempting AAV9 mediated transduction of organoids.

Figure 3and 6:
RO and RPE transduction was already done by Gonzalez codero 2018 (Citation 19) exceptt for 7m8.
Please explain the novelty and impact

Response 4.

We are grateful to the reviewer for highlighting that the novelty of our work was not sufficiently clear. To begin with, our protocol of differentiation is completely different from the one used by Gonzalez-Cordero et al 2018, Our protocol produces cone-enriched retinal organoids instead of mature photoreceptor phenotypes. Our work also demonstrates for the first time that photoreceptors can be targeted with different AAV capsids with different tropisms at the neural retina stage. For the first time AAV cell surface receptors and co-receptors were identified in the retinal organoids at the timepoint of infection. Moreover, our work correlates the presence of AAV receptors on the day of transduction with the efficiency of gene delivery. For the first time AAVR, initially described as a universal AAV receptor is shown in retinal organoids. Also, AAV9 has a natural tropism towards mature cones but despite of this, it did not successfully target all the cones in earlier stages or in other organoids (Welby 2017 and Gonzalez Cordero 2018). For the first time we propose the absence of N-linked galactose in retinal organoids until very late stages as a putative mechanism explaining this observation. We have made corrections in the manuscript to highlight all these novelties

Figure 5: is doubled

Response 5. Thank you for highlighting this error. By mistake Fig 5 was duplicated. This has been rectified in the revised manuscript

Round 2

Reviewer 2 Report

The manuscript of Garita-Hernandez et al. deals with the viral transduction of genes to 3D retinal organoids. While the manuscript is quite well written and pictures are of good quality, the manuscript needs improvement.

Line 152/3:
Expression of Lam and Syndecan at day70:
No proof there is a correlation between that and the 7m8 efficacy. Could they do a knowdown or inhibitoin?

Line 156:
One of the major aspects (AAV2 vs 7m8) is just referred to a hypothesis from a publication

Line 191:
AAV9
-->PNA lectin only present at day 226
Why not test infection at older stages? (Will also not work, so no proof for a correlation here)

Figure 3 and 6:
RO and RPE transduction was already done by Gonzalez codero 2018 (Citation 19) expcet for 7m8.
Please discuss the novelty here

Figure 5: double in the PDF

Author Response

Response to Reviewer 2 Comments Round 2

The manuscript by Garita-Hernandez et al. describes AAV-mediated gene delivery to retinal organoids of different age. While the depiction of the results is nice, there is several concerns to the claims of the authors:

We appreciate the revision performed by Reviewer 2 and the opportunity to improve our manuscript and to put in value our results. We just want to clarify that out study focus in early transduction of retinal organoids and we have made some changes in the manuscript to highlight this as it was not sufficiently clear in the submitted version.

We respond in detail to the reviewer's concerns below.

Line 152/3:
Expression of Lam and Syndecan at day70:
No proof there is a correlation between that and the 7m8 efficacy. Could they do a knowdown or inhibition?
Line 156: One of the major aspects (AAV2 vs 7m8) is just referred to a hypothesis from a publication

Response 1 and 2. Indeed, we have not done any knockdown experiments to support this claim and unfortunately, we are not in a position to do these experiments as these would entail significant modifications on the behavior of the cells and their differentiation into organoids if done at the iPS stage and delivery of siRNA in differentiated organoids would necessitate additional protocols of delivery which we have not established in the laboratory. However, previous experiments already probed glycan dependencies of 7m8 compared to AAV2 (see supplementary figure 1 Dalkara et al., 2013). In this prior work ex vivo characterization of 7m8 on relevant cell types was performed compared to AAV2 on CHO, PgsA, Pro5, and Lec1 cells. CHO/PgsA transduction probes the HSPG dependence, whereas Pro5/Lec1 transduction examines the sialic acid dependence, of AAV2 and 7m8. Furthermore, heparin binding profiles of 7m8 and AAV2 were assessed by heparin column elution profile by chromatography, thus supporting experimentally the hypothesis from previous publication.

Line 191:
AAV9
-->PNA lectin only present at day 226
Why not test infection at older stages? (Will also not work, so no proof for a correlation here)

Response 3. The aim of this paper is to compare the efficiency of transduction by different AAV capsids early in differentiation. AAV9 receptor, N-galactose, identified by PNA lectin is not present in our organoids at the time of infection (D44) nor at the end of the experiments (D70). The absence of PNA implies this is the reason for the lack of expression observed with AAV9. However, we did show the presence of PNA in mature organoids (D226) using our protocol of differentiation, and we suggest the time at which PNA becomes available will vary with the protocol of differentiation used. Looking for the presence of the receptor in the organoids before attempting AAV-mediated transduction is thus proposed for readers attempting AAV9 mediated transduction of organoids.

Figure 3 and 6:
RO and RPE transduction was already done by Gonzalez codero 2018 (Citation 19) expcet for 7m8.
Please discuss the novelty here

Response 4.

We are grateful to the reviewer for highlighting that the novelty of our work was not sufficiently clear. To begin with, our protocol of differentiation is completely different from the one used by Gonzalez-Cordero et al 2018, Our protocol produces cone-enriched retinal organoids instead of mature photoreceptor phenotypes. Our work also demonstrates for the first time that photoreceptors can be targeted with different AAV capsids with different tropisms at the neural retina stage, when it is desirable to perform transformations of retinal organoids prior to transplantation studies. Also, for the first time AAV cell surface receptors and co-receptors were identified in the retinal organoids at the timepoint of infection. Moreover, our work correlates the presence of AAV receptors on the day of transduction with the efficiency of gene delivery. For the first time AAVR, initially described as a universal AAV receptor is shown in retinal organoids. Also, AAV9 has a natural tropism towards mature cones but despite of this, it did not successfully target all the cones in earlier stages or in other organoids (Welby 2017 and Gonzalez Cordero 2018). For the first time we propose the absence of N-linked galactose in retinal organoids until very late stages as a putative mechanism explaining this observation. We have made corrections in the manuscript to highlight all these novelties.

Figure 5: double in the PDF

Response 5. This error has been rectified in the revised manuscript submitted in the first round of revisions